THE NATURAL HISTORY OF MODEL ORGANISMS

# The house sparrow in the service of basic and applied biology

**Abstract** From the northernmost tip of Scandinavia to the southernmost corner of Patagonia, and across six continents, house sparrows (*Passer domesticus*) inhabit most human-modified habitats of the globe. With over 7,000 articles published, the species has become a workhorse for not only the study of self-urbanized wildlife, but also for understanding life history and body size evolution, sexual selection and many other biological phenomena. Traditionally, house sparrows were studied for their adaptations to local biotic and climatic conditions, but more recently, the species has come to serve as a focus for studies seeking to reveal the genomic, epigenetic and physiological underpinnings of success among invasive vertebrate species. Here, we review the natural history of house sparrows, highlight what the study of these birds has meant to bioscience generally, and describe the many resources available for future work on this species.

**HALEY E HANSON\*, NOREEN S MATHEWS, MARK E HAUBER AND LYNN B MARTIN**

**\*For correspondence:**
haleyehanson@gmail.com

**Competing interests:** The authors declare that no competing interests exist.

## Introduction

House sparrows are small, sexually dimorphic birds in the family Passeridae. The species is one of the most widely distributed and common birds in the world, represented by 12 different subspecies (*Summers-Smith, 2009*). House sparrows can be found living and breeding in climactically extreme environments from deserts in southern California to cities above the Arctic circle, where they are found almost exclusively in close proximity to human habitation (*Hanson et al., 2020b*). Considered anthrodependent, some populations have gone extinct locally without human presence (*Ravinet et al., 2018*; *Summers-Smith, 1988*). It is for this relationship with people that they received their species identifier *domesticus*, which derives from the Latin *domus* or 'house', from Carl Linnaeus in 1758 (*Jobling, 2009*; *Anderson, 2006*). Their ubiquity and close association with humans have undoubtedly led to their detailed study across biological and even sociological disciplines. Here, we explore the natural history of

house sparrows and the contributions that these birds have made to basic biology and beyond.

## Native distribution and natural range expansions

House sparrows are native to parts of Asia, North Africa and most of Europe, (with the exception of Italy which is occupied by the Italian sparrow *P. italiae*; *Animation 1*). Becoming commensal some 10,000 years ago, house sparrows are now strongly associated with habitats that have been modified by humans. However, they also continue to increase their geographic range by exploiting ongoing and accelerating anthropogenic change (*Ravinet et al., 2018*; *Saetre et al., 2012*). A reliance on humans is evident from their colonization of Northern Europe, Eastern Europe and Central Asia in the early 1800s, as agriculture spread and urbanization increased (*Summers-Smith, 1963*). Though still widespread, significant declines have been reported in the native range of the species since the 1970s. This topic remains contentious (*Box 1*), but these declines have been attributed

**Figure 1.** Adult and nestling house sparrows. (A) Female house sparrow. (B) Male house sparrow. (C) Nestling house sparrows. (D) Male house sparrow provisioning nestlings. Image Credits: All images taken by Janneke Case in Tampa, Florida, United States, in 2019.

to a multitude of factors, including infectious disease, pollution, pesticide use, predator dynamics, new building methodologies and more efficient grain harvesting and storage (*Shaw et al., 2008*; *Summers-Smith, 2003*; *Singh et al., 2013*; *Bell et al., 2010*; *Dadam et al., 2019*).

## Introduced distribution and range expansions

House sparrows are one of the most ubiquitous birds in the world (*Anderson, 2006*). In approximately 170 years, they colonized the globe such that they now reside in every continent except Antarctica and occupy an estimated 76,600,000 km$^2$ (*Birdlife international, 2018*). There have been over 250 introduction or translocation events recorded worldwide (*Table 1*), with the first deliberate successful introduction occurring in 1851 in New York City (*Summers-Smith, 1988*). Many introductions stemmed from colonial acclimatization societies purposefully releasing birds for cultural reasons or as failed attempts at biological control. More recently, introductions have been accidental. Ship-assisted dispersal (e.g., cargo ships, cruise liners) has been documented, and other types of vehicle-assisted dispersal are also likely (*Sainz-Borgo et al., 2016*; *Schrey et al., 2014*; *Clergeau et al., 2004*; *Szent-Ivány, 1959*; *Summers-Smith, 1963*).

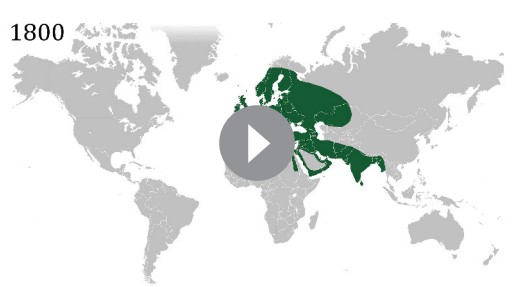

1800

**Animation 1.** House sparrow distribution from 1800 to 2019.

https://elifesciences.org/articles/52803#video1

Image Credit: Haley E Hanson, Noreen S Mathews, and Jaime E Zolik. For sources used, please refer to https://doi.org/10.6084/m9.figshare.11915955.v1.

## Dimorphism in morphology and behavior

Male house sparrows tend to be heavier and larger than females (*Figure 1*; *Hanson et al., 2020b*). Plumage coloration differs between the sexes. Males have gray crests and black postocular stripes with conspicuous white spots behind the eyes (*Figure 1b*). Male abdomens are gray whereas bills, tails, wings and body feathers are black or dark brown. Plumage in females is drabber, with crests that are dark brown and post-ocular stripes that are light brown. Females lack black head markings and have gray-brown to light brown cheeks, bills and feathers (*Figure 1a*). Female plumage resembles juveniles and females from other *Passer* species so much that distinguishing them visually is often difficult (*Anderson, 2006*). Subspecies also differ in size, mass and male plumage (See *Summers-Smith, 1988*).

The most conspicuous morphological difference between male and female sparrows is the large black throat badge of males. Arguably, this badge is one of the factors that made this species a model in behavioral ecology (*Sánchez-Tójar et al., 2018*). Large badge size has been thought to convey an individual's propensity to win in male-male competitive interactions; the logic was that possessing information *a priori* about a competitor could save both the badge-holder and his opponents from wasted energy and risk of injury (*Rohwer, 1975*). Recently, however, the largest meta-analysis to date revealed that badge size is at best an unreliable signal of dominance status (*Sánchez-Tójar et al., 2018*). The currently favored hypothesis for badge size is that it serves some role in mate choice, as females tend to choose males with large badges, and badge size is positively correlated with male sexual behaviors (*Veiga, 1996*).

Importantly, many morphological characteristics also vary geographically. Most well-known through the pioneering work of Richard F Johnston and Robert K Selander, plumage color and aspects of body size (wing, tail and tarsus length, skeletal characteristics, and body mass) were found to vary within and between native and introduced populations (*Selander and Johnston, 1967*; *Johnston and Selander, 1964*; *Johnston and Selander, 1971*; *Johnston, 1969*; *Johnston, 1973*). Introduced populations in North America were discovered to have pale coloration in hot, arid climates, but darker coloration in cooler, humid climates (*Johnston and Selander, 1964*). Body size of birds also increased with latitude, and perhaps most interestingly, all of these geographic trends in biological traits arose rapidly in the introduced populations (*Johnston and Selander, 1964*; *Selander and Johnston, 1967*; *Johnston and Selander, 1971*).

## Diet and foraging

Nestling house sparrows are fed an insect-based diet for the first three days after hatching. Later, following fledging, they favor grains, especially outside urban areas (*Anderson, 2006*). Adult house sparrows have a fairly opportunistic diet throughout much of the year, especially in cities and suburbs where human refuse is plentiful (*Summers-Smith, 1988*). One of the reasons house sparrows are so adept at exploiting diverse diets might involve plasticity in the release of digestive enzymes (*Brzek et al., 2009*). Behaviorally, responses to food also seem to play a role in range expansions, another reason this species has been used as a model. For example, house sparrows in the roughly 40-year-old Panama population consume unfamiliar foods more quickly than birds from a much older invasive population in New Jersey in the United States (*Martin and Fitzgerald, 2005*). A similar pattern is observed among Kenyan sparrows such that birds living at the expanding range edge of that colonization approach and eat novel foods more quickly than birds from the core of the population (*Liebl and Martin, 2014*).

A tendency to eat novel foods may benefit birds in habitats where resources are scarce or unfamiliar, but such behavior could also come with risks. Spoiled foods or exposure to novel toxins, for example, may activate the immune system (*Martin and Fitzgerald, 2005*). This

<div style="background:lightblue">

## Box 1. Outstanding questions about the natural history of house sparrows.

1. How did house sparrows come to colonize most of the planet? What characteristics make them more successful than most vertebrate species? Are some populations or subspecies more predisposed to invading new areas than others?

2. How do house sparrows cope with the apparent challenges of urban life such as light, noise and air pollution?

3. What factors are contributing to the decline of house sparrow populations worldwide (both in native and introduced populations), and are these bellwethers for the impending decline of phylogenetically and/or ecologically related species?

</div>

notion is supported by the observation that populations differ quite extensively in how their immune systems are organized and what parasites they harbor throughout their lives (*Kilvitis et al., 2019*; *Martin et al., 2015*; *Martin et al., 2014*; *Coon and Martin, 2014*; *Coon et al., 2014*).

## Breeding biology

Sparrows tend to build nests in pre-existing cavities, but they also routinely nest in roofs, eaves and walls of human-built structures (*Figure 1c*) as well as in densely branched trees and shrubs (*Anderson, 2006*; *Sheldon and Griffith, 2017*; *Manna et al., 2017*). Nests are comprised mostly of vegetation but some clay, sand, cloth and even dung may be used (*Heij, 1986*). In some cities, nests also contain aromatic plants or even cigarette butts that contain antiparasitic secondary compounds (*Sengupta and Shrilata, 1997*). Males initially choose nesting sites and subsequently advertise for mates by vocal and visual displays (*Summers-Smith, 1963*). However, unlike many songbirds, males exhibit aggressive, territorial behavior only in a very small area around the nest site. Females select males based on visual and vocal displays and the location of nest sites (*Anderson, 2006*). Once paired, males and females often remain together for the entire season or even multiple years. Pairs also commonly use the same nest site for several years (*Summers-Smith, 1963*), however, as is typical in most bird species, males are more likely to stay in, or habitually return to, the area around a nest site than females (*Morrison et al., 2008*). Both sexes defend the nest, brood the eggs and care for the young, though females put more effort into the brooding than males (*Figure 1d*; *Anderson, 2006*). Pairs are socially monogamous, however, the proportion of offspring that are fathered by an extra-pair male (extra-pair paternity) can reach 26%, particularly if food is scarce and the environment is harsh (*Stewart et al., 2015*). House sparrows typically begin breeding during the first year of life, but breeding success is comparatively low in younger breeders (*Hatch and Westneat, 2007*).

**Table 1.** Global house sparrow introduction or translocation events by region.
Introduction and translocation events include both purposeful and inadvertent release of any number of birds from all subspecies, successful or unsuccessful. We list a range instead of a single number because of discrepancies among published reports. For sources used, please refer to https://doi.org/10.6084/m9.figshare.11915955.v1.

| Region | Number of introductions or translocations |
| --- | --- |
| Africa | 24–43 |
| Asia | 9–11 |
| Oceania | 54–60 |
| Europe | 4+ |
| North America | 135–136 |
| South America | 32–35+ |

Reproductive biology has been another reason this species has been used as a model, in particular to understand the cues that influence the onset of breeding. Towards the global poles, house sparrows, like other species, rely on changes in the number of hours of daylight and temperature to ensure that breeding coincides with peak food availability (*Hau, 2001*). Nearer to the equator, however, both light levels and temperature are fairly stable year-round (*Hau et al., 1998*), and house sparrows in this region seem to use changes in precipitation regimes to time breeding. In Panama, India and Malawi, for instance, house sparrows breed predominantly during the dry parts of the year, but in Zambia, sparrows breed both five months prior to the peak of the rains, and again when the rains are ongoing (*Nhlane, 2000*; *Hanson et al., 2020b*).

Perhaps the main reason that house sparrows have been a model organism in basic ornithology involves the variation they show in life history and associated physiological traits along gradients in their geographic range. Known as clinal variation, in house sparrows, this phenomenon has been documented for metabolic rates (*Hudson and Kimzey, 1966*; *Kendeigh and Blem, 1974*; *Blem, 1973*), hormone regulation (*Romero et al., 2006*; *Breuner and Orchinik, 2001*; *Liebl and Martin, 2012*), and immune defenses (*Kilvitis et al., 2019*; *Martin and Fitzgerald, 2005*; *Martin et al., 2004*). These trends are best-reflected by clinal variation in clutch size; just as in most songbirds, house sparrow clutches are small near the equator and increase pole-ward (*Anderson, 2006*). This pattern, which exists in both the native and non-native distribution, is intriguing because of the recency of most introductions. Such recency means that new populations would have had little time for genetic adaptation as well as being exposed to founder effects and other genetic challenges (i. e., bottlenecks) inherent to introductions (*Baker, 1995*; *Lowther, 1977*).

## Genetics, epigenetics and the microbiome

Given the broad distribution of the species and its recent arrival in many regions, house sparrows have been used as models of genetic, genomic and more recently epigenetic changes during range expansion. Early studies using allozymes (variants of enzymes encoded by alleles of the same gene) revealed little genetic variation among and within North American populations, but suggested that introduced populations underwent genetic bottlenecks and were significantly differentiated from source and native European populations (*Parkin and Cole, 1985*; *Klitz, 1973*). DNA fingerprinting, or minisatellites, was used on house sparrows before any other bird species, and microsatellite research followed soon after, revealing subtler genetic differences among populations (*Burke and Bruford, 1987*; *Neumann and Wetton, 1996*). Microsatellite analyses have been valuable to inferring invasion history, population structure and dispersal behavior, as well as establishing relatedness such as parentage (*Wetzel et al., 2012*; *Mock et al., 2012*; *Schroeder et al., 2013*; *Jensen et al., 2013*; *Liker et al., 2009*; *Schrey et al., 2011*; *Lima et al., 2012*; *Schrey et al., 2014*; *Andrew et al., 2018b*; *Kekkonen et al., 2011*). Critically, it was microsatellite data that provided the genetic evidence of extra-pair paternity in this socially monogamous, pair-bonded species (*Griffith et al., 1999*).

Recently, an annotated genome became available for house sparrows (*Elgvin et al., 2017*). The genome belongs to a female house sparrow from a pedigreed, inbred population from the island of Aldra in Norway, and was studied to better understand speciation in the Italian sparrow (*P. italiae*; *Elgvin et al., 2017*). The Italian sparrow is a hybrid of the house sparrow and the Spanish sparrow (*P. hispaniolensis*), and this system has led to a wealth of insight about genetic mechanisms affecting hybrid speciation (*Hermansen et al., 2011*; *Hermansen et al., 2014*; *Trier et al., 2014*; *Elgvin et al., 2017*; *Elgvin et al., 2011*). For example, *Runemark et al. (2018)* investigated the genomes of isolated island populations of the Italian sparrow to understand the formation of hybrid genomes. They found that the contribution of parental genome (in this case, the house sparrow and the Spanish sparrow) can differ greatly across populations, but some genomic regions have less variation than others.

Prior to the annotated genome, a high-density single-nucleotide polymorphism (SNP) array was developed for the species (*Hagen et al., 2013*; *Lundregan et al., 2018*). This tool was used to detect signatures of adaptation in introduced populations in climatically varied environments across Australia, and to understand the genetic basis of variation in bill morphology (*Andrew et al., 2018a*; *Lundregan et al., 2018*). Other next-generation sequencing tools, such as tissue-specific transcriptomic assemblies, a

seminal fluid proteome and a genome-wide linkage map, are also available for the species (*Ekblom et al., 2014*; *Razali et al., 2017*; *Matsushima et al., 2019*; *Mirón et al., 2014*; *Rowe et al., 2020*; *Hagen et al., 2020*).

Epigenetic variation, namely DNA methylation, has also begun to be investigated in house sparrows (*Kilvitis et al., 2018*; *Kilvitis et al., 2019*; *Riyahi et al., 2017*). House sparrows exhibit marked phenotypic variation across introduced populations, even though many non-native populations experienced bottlenecks and founder effects upon introduction (*Johnston and Selander, 1971*; *Liebl and Martin, 2012*; *Martin et al., 2015*; *Bókony et al., 2012*; *Martin and Fitzgerald, 2005*; *Ben Cohen and Dor, 2018*; *Hanson et al., 2020b*). It has been hypothesized that DNA methylation or other molecular epigenetic mechanisms may have affected the ability of populations to colonize new areas (*Box 1*). *Schrey et al. (2012)*, for example, found that variation in DNA methylation was inversely correlated with genetic diversity among recently invaded Kenyan populations, suggesting that populations might compensate for low genetic diversity with epigenetic diversity. In Australian house sparrows, a similar pattern was found as well as an epigenetic signature mirroring that of genetic population clustering arising from the original source population (*Sheldon et al., 2018*). These observations and other data led to the hypothesis that house sparrows might exhibit high epigenetic potential, or the capacity for epigenetic mechanisms within the genome to facilitate phenotypic plasticity (*Kilvitis et al., 2017*). One form of epigenetic potential is the number of CpG sites (sequences in the genome where DNA methylation can occur) in gene promoters. Indeed, towards the expanding edge of the very recent Kenyan invasion, CpG sites across the genome were significantly higher than in older Kenyan house sparrow populations, suggesting that epigenetic potential may generally mediate the introduction success of the species (*Hanson et al., 2020a*).

In addition to epigenetic mechanisms, the microbiome could also play an important role in the ecology of the species (*Russell et al., 2012*; *Borre et al., 2014*). Gut microbes affect the growth rates of house sparrow nestlings (*Kohl et al., 2018*), and nestlings and adults differ in the structure and membership of their microbial communities, with the nestling microbial community being affected by social and genetic family affiliation but also diet and environmental microbes (*Kohl et al., 2019*). Further studies are needed to understand what the microbiome means to the house sparrow, particularly as this bird favors the same areas as humans.

As new technologies are developed and refined, we expect the interest in house sparrow genetics, epigenetics and the microbiome to grow. Several local populations of house sparrows have been pedigreed, which enables quantitative genetic estimates of heritability and genetic architecture (*Schroeder et al., 2015*; *Jensen et al., 2003*; *Wetzel et al., 2012*). Additionally, many museums have large collections of house sparrows including many specimens collected before 1900 (*Table 2*). These collections will be valuable sources of genetic and morphologic data, as well as for use in analyses of pollutants during different eras of human cohabitation (e.g., *DuBay and Fuldner, 2017*).

## Conclusions

Advocating that house sparrows be used as model organisms is not simple as many definitions of model species are available (*Bolker, 2009*; *Bolker, 2014*; *Bolker, 2017*). This jumble of definitions has led some to claim that 'model' is one of the most under-powered concepts in biology (*Katz, 2016*). These challenges motivated us to think hard about how house sparrows could serve as models (*Bolker, 2009*). Besides their historic value in the contexts discussed above (i.e., invasion genetics, behavioral ecology, life history evolution), we feel that they generally promise a high return in basic, practical and even economic insight, a value not attributable to many other species.

Previously, *Bedford and Hoekstra (2015)* made a form of this argument about the mouse genus *Peromyscus*. Specifically, they cast the enormous amount of information available for *Peromyscus* as ideal for modelling intraspecific variation. We are skeptical whether any species can really model variation; there are simply too many interactions possible within genomes, not to mention disparities in the forms and forces of selection and plasticity among populations. We agree, though, that *Peromyscus,* house sparrows and probably other species could be representative for many small, short-lived and broadly distributed vertebrates that are benefitting from human activity (e.g., urbanization). Moreover, as with *Peromyscus* species for Lyme disease, Hantavirus and other zoonoses, house sparrows play important roles in local infectious disease

**Table 2.** House sparrows available in museum collections.
Listed are the five largest house sparrow museum collections, the number of specimens present in each and the time of specimen sampling. Data was compiled from all collections present in the Vert-Net database (*Constable et al., 2010*). For search terms and the full table, please refer to https://doi.org/10.6084/m9.figshare.1915955.v1.

| Collection | Number of specimens |
| --- | --- |
| University of Kansas Biodiversity Institute (KU) | 12,830 |
| Royal Ontario Museum (ROM) | 7,654 |
| Field Museum of Natural History (FMNH) | 1,974 |
| Museum of Vertebrate Zoology, UC Berkeley (MVZ) | 1,888 |
| American Museum of Natural History (AMNH) | 1,776 |
|  |  |
| Specimens collected before 1900 | 1,597 |
| Specimens collected between 1900–1950 | 7,460 |
| Specimens collected after 1950 | 29,401 |

risk, including West Nile virus, *Salmonella* and other infections (*Ostfeld et al., 2014*; *Tizard, 2004*; *Kilpatrick et al., 2007*).

Furthermore, although we and others have tended to focus on them as an exemplary invader, house sparrows also promise insight into the range expansions and contractions of native species, phenomena becoming more common as the global climate continues to change (*Box 1*). Just like George Box's claim for mathematical models, no model organism is perfect, but many can be informative (*Bolker, 2014*; *Box, 1976*). Although all model organisms will thus have some shortcomings, some, such as the house sparrow, might provide unique value by helping us learn how to mitigate anthropogenic effects on natural areas and systems (*Manger, 2008*).

### Acknowledgements
We thank Janneke Case for allowing us to use her photographs, and Jaime Zolik for her help in the production of the distribution animation.

**Haley E Hanson** is in the Global and Planetary Health strategic area at the University of South Florida, Tampa, United States
haleyehanson@gmail.com
https://orcid.org/0000-0002-0513-5911

**Noreen S Mathews** is in the Global and Planetary Health strategic area at the University of South Florida, Tampa, United States

**Mark E Hauber** is in the Department of Evolution, Ecology and Behavior, School of Integrative Biology, University of Illinois at Urbana-Champaign, Urbana and Champaign, United States

**Lynn B Martin** is in the Global and Planetary Health strategic area at the University of South Florida, Tampa, United States

*Author contributions:* Haley E Hanson, Visualization, Writing - original draft, Writing - review and editing; Noreen S Mathews, Visualization, Writing - review and editing; Mark E Hauber, Conceptualization, Writing - original draft, Writing - review and editing; Lynn B Martin, Conceptualization, Supervision, Writing - original draft, Writing - review and editing

*Competing interests:* The authors declare that no competing interests exist.

### Funding

| Funder | Grant reference number | Author |
| --- | --- | --- |
| University of South Florida College of Public Health |  | Lynn B Martin |
| Sigma Xi | G2016100191872782 | Haley E Hanson |
| Porter Family Foundation |  | Haley E Hanson |
| American Ornithological Society | Hesse Grant | Haley E Hanson |
| American Museum of Natural History | Frank M. Chapman Memorial Fund | Haley E Hanson |
| United States-Israel Binational Science Foundation | 2017258 | Mark E Hauber |

The funders had no role in study design, data collection and interpretation, or the decision to submit the work for publication.

## Decision letter and Author response

Decision letter https://doi.org/10.7554/eLife.52803.sa1
Author response https://doi.org/10.7554/eLife.52803.sa2

## Additional files

### Data availability

All data generated during the preparation of this review are included in the manuscript and the supplementary information is available on figshare (https://doi.org/10.6084/m9.figshare.11915955.v1).

The following dataset was generated:

| Author(s) | Year | Dataset URL | Database and Identifier |
|---|---|---|---|
| Hanson HE, Matthews NS, Hauber ME, Martin LB | 2020 | https://doi.org/10.6084/m9.figshare.11915955.v1 | figshare, 10.6084/m9.figshare.11915955.v1 |

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
