## [Decision Letter]

**Acceptance summary:**

The reviewers felt that this was a well-written and entertaining short review on a study species for which "there is surprisingly little general literature". We believe that the article will make a strong contribution to "The Natural History of Model Organisms" collection.

**Decision letter after peer review:**

Thank you for submitting your article for consideration by *eLife*. Your article has been favourably reviewed by two peer reviewers, and the evaluation has been overseen by Stuart King as the Associate Features Editor. The following individual involved in review of your submission has agreed to reveal their identity: Mark Ravinet (Reviewer #1). The editor has drafted this decision to help you prepare a revised submission.

Summary:

This essay is being considered as part of a series of articles on "The Natural History of Model Organisms" (https://elifesciences.org/collections/8de90445/the-natural-history-of-model-organisms). Each article should explain how our knowledge of the natural history of a model organism has informed recent advances in biology, and how understanding its natural history can influence/advance future studies.

This specific article provides a summary of our current knowledge on the biology and evolution of the house sparrow, and discusses its place as a model species for biological insight. Both reviewers were very positive about the article, finding it to be a well-written and entertaining short review on a study species for which "there is surprisingly little general literature (with the exception of a few rather niche books)". Reviewer 1 noted, "I learned quite a bit and I am grateful to the authors for pointing me towards some useful references I should add to my reading list".

Neither reviewer had any major concerns, but they did make a few suggestions that could strengthen the article. These mostly relate to refining the focus of the text, while one relates to the tables included in the article.

Suggested revisions:

1) Invasion biology

Since, invasion biology distinguishes the house sparrow from the other model organisms included in the *eLife* collection thus far, the reviewers wondered if the text could be reorganized slightly to make this theme more prominent. The general feeling was that this could be achieved by simply renaming some of the sections and adding a few guiding sentences throughout.

2) Epigenetic basis of phenotypic plasticity

The article gives a lot of detail on the epigenetic basis of phenotypic plasticity in house sparrows. This is not that surprising given a lot of the work on the system is done by the authors. The reviewers, however, felt it would be better to balance this with a bit more on the recent evolutionary genomics on the species. For example, the Italian sparrow is mentioned only in passing but it is quite an important aspect of why this study system is so interesting to speciation research. Work by Elgvin et al., 2017 and 2011, Hermansen et al., 2014, and Trier et al., 2014 is worth looking at.

3) Conclusion

The reviewers thought the conclusion would be stronger if it simply focused on the strengths of the house sparrow as a study organism, and the questions that may be best answered with this species in the future. The sentences agreeing and disagreeing with the Peromyscus article should probably be removed. Different model organisms are best suited for answering different questions, and all model organisms have shortcomings.

4) Tables

Please add references to both tables make it clear where the data came from. This could be done by adding an extra column to each table for the relevant citations. Alternatively, if you opt to upload the original datasets you compiled (and related references) to a repository such as Figshare, you could simply cite the dataset in the captions for the tables. Please also provide more detail about Table 2 to make it clear whether your search was focused on North American collections, or whether the search was more widespread and all the collections just happen to be in North America by chance.

---

## [Author Response]

Suggested revisions:1) Invasion biologySince, invasion biology distinguishes the house sparrow from the other model organisms included in the eLife collection thus far, the reviewers wondered if the text could be reorganized slightly to make this theme more prominent. The general feeling was that this could be achieved by simply renaming some of the sections and adding a few guiding sentences throughout.

We agree that the house sparrow is an extraordinarily successful invasive species, and we highlight this aspect of its natural history throughout the manuscript. But to exclusively discuss this aspect only may detract from other elements of the natural history, behavior, morphology, and physiology that have also been important (e.g., badges as ornaments, biorhythms, geographic variation in metabolic rates and body size). For these reasons, we chose not to reorganize.

2) Epigenetic basis of phenotypic plasticityThe article gives a lot of detail on the epigenetic basis of phenotypic plasticity in house sparrows. This is not that surprising given a lot of the work on the system is done by the authors. The reviewers, however, felt it would be better to balance this with a bit more on the recent evolutionary genomics on the species. For example, the Italian sparrow is mentioned only in passing but it is quite an important aspect of why this study system is so interesting to speciation research. Work by Elgvin et al., 2017 and 2011, Hermansen et al., 2014, and Trier et al., 2014 is worth looking at.

We agree and have included an additional paragraph about hybrid speciation of the Italian sparrow including the requested citations. We consider to have now balanced the genomic elements with the epigenetic elements in our discussion.

3) ConclusionThe reviewers thought the conclusion would be stronger if it simply focused on the strengths of the house sparrow as a study organism, and the questions that may be best answered with this species in the future. The sentences agreeing and disagreeing with the Peromyscus article should probably be removed. Different model organisms are best suited for answering different questions, and all model organisms have shortcomings.

We agree that all model organisms have shortcomings, and indeed, we worry that the concept of model organism has taken on so many meanings recently that the term has started to lose its utility. To make this case and emphasize the relevant value of house sparrows as a particular type of model, we feel that we need to juxtapose it with more traditional model organisms. This approach also justifies important areas of future study for our focal species.

4) TablesPlease add references to both tables make it clear where the data came from. This could be done by adding an extra column to each table for the relevant citations. Alternatively, if you opt to upload the original datasets you compiled (and related references) to a repository such as Figshare, you could simply cite the dataset in the captions for the tables. Please also provide more detail about Table 2 to make it clear whether your search was focused on North American collections, or whether the search was more widespread and all the collections just happen to be in North America by chance.

We have made the sources used to produce the figures and the tables available on Figshare with the DOI embedded in the caption. Further, we made the extended version of Table 2 available and included the date of the search and all search terms used to compile the table. We also updated the caption to reflect that all institutions with data in the VertNet database were included in the production of the table. We updated Table 2, as well.